# Spatiotemporal Ultrastructural, Histological, and Morphometric Changes in the Buccal Cavity of Grass Carp (*Ctenopharyngodon idella*) During Fingerling, Yearling, and Adult Stages

**DOI:** 10.3390/ani14213162

**Published:** 2024-11-04

**Authors:** Ahmed M. Abdellatif, Ahmed I. Ateya, Khadiga A. Hasan, Mansour A. Alghamdi, Fatma A. Madkour

**Affiliations:** 1Department of Anatomy and Embryology, Faculty of Veterinary Medicine, Mansoura University, Mansoura 35516, Egypt; 2Department of Development of Animal Wealth, Faculty of Veterinary Medicine, Mansoura University, Mansoura 35516, Egypt; dr_ahmedismail@mans.edu.eg; 3Botany Department, Faculty of Science, Mansoura University, Mansoura 35516, Egypt; khadiga.ahmed.php@gmail.com; 4Electron Microscopy Unit, Mansura University, Mansoura 35516, Egypt; 5Department of Anatomy, College of Medicine, King Khalid University, Abha 62529, Saudi Arabia; m.alghamdi@kku.edu.sa; 6Genomics and Personalized Medicine Unit, The Center for Medical and Health Research, King Khalid University, Abha 62529, Saudi Arabia; 7Department of Anatomy and Embryology, Faculty of Veterinary Medicine, South Valley University, Qena 83523, Egypt; madkour.f@vet.svu.edu.eg

**Keywords:** Cyprinidae, digestive tract, goblet cell, jaw, lip, mouth, taste bud, teleost, tongue, ultrastructure, vertebrates

## Abstract

The grass carp, also called *Ctenopharyngodon idella*, is a fish of aquacultural and research importance. In the present study, different parts of the buccal cavity in *C. idella* were studied using light and scanning electron microscopes. Three developmental stages were selected, as follows: fingerling (three months posthatching, mph), yearling (12 mph), and adult (48 mph). The jaw epithelium was the thickest part of the buccal epithelium and appeared smooth in fingerlings but roughened in yearlings. In adult fish, the jaw epithelium was arranged into dome-shaped cellular masses separated from each other by narrow grooves. Each mass consists of many keratinocytes with surface wrinkles. No taste buds—microscopic chemosensory organs—were seen in the jaw mucosa. However, many taste buds and mucus-secreting cells were observed throughout the mucosa of the rest of the buccal cavity, concentrated within the roof and floor of the buccal cavity, suggesting a role for these parts in food tasting and mouth lubrication. This study is the first to reveal maturational changes in the buccal cavity of grass carp and will serve as a basis for the interpretation of diseases affecting the buccal cavity of fish.

## 1. Introduction

The buccal cavity, or mouth, of vertebrates represents the foremost segment of the digestive tract. It is lined by stratified squamous epithelium with abundant goblet cells [1]. In teleost, the buccal cavity is bounded rostrally by the upper and lower lips, dorsally by the palate, and ventrally by the oral floor. The tongue of several teleost species is poorly developed, lacks definitive muscles, and appears mainly as a fold in the oral floor. In most species, an additional structure termed the oral valve, or the oral breathing valve, is located caudal to the fish jaw and is believed to play a crucial role in mouth closure during blood oxygenation in gills [2].

The appearance of the mouth varies among species and may reflect the digestive behavior of fish [3]. For instance, in typical carnivores, e.g., sharks, the mouth appears broad, ventrally positioned, and equipped with numerous sharp teeth, while in herbivores, e.g., cyprinids, the mouth is terminally located, less expansive, and devoid of teeth. It is noteworthy that in most cyprinids, the process of food trituration is mainly achieved by the action of well-developed pharyngeal teeth that were found to display age-related morphologic and ultrastructural conformations [4]. In Siluridae, a family of catfish of well-known omnivorous behavior, the mouth is wide and fringed with sensitive barbels that enable them to seek and sort food particles of mixed nature in muddy areas [5].

The buccal cavity performs crucial roles related to food discrimination and assortment. The ability of fish to recognize food particles of various tastes and textures is poorly understood; however, it could be attributed to the presence of a large number of chemoreceptive structures, the taste buds, within the epithelium of the buccal cavity. The number and pattern of arrangement of these buds are modulated by many factors, including fish species, age, and nature of the surrounding environment [6]. For example, in amago salmon (*Oncorhynchus rhodurus*), the epithelium of the buccal cavity contained a limited number of taste buds during the first two months of posthatching life; however, a sharp increase in their numbers was seen during the subsequent few months coincident with gaining the ability of juvenile fish to migrate from freshwater to seawater [7].

The grass carp (*Ctenopharyngodon idella*) is a Cyprinidae member that was initially native to Eastern Asia, from which it spread to various regions of the world [8]. *C. idella* has the ability to consume huge amounts of submerged plants and is thus widely employed as a green tool for the eradication of aquatic weeds [9]. Moreover, it has been established as a model for testing several digestive diseases [10]. Furthermore, biological materials derived from *C. idella* have been proven to be beneficial for wound healing and tissue regeneration [11].

*C. idella* reaches sexual maturity relatively late, around the third to fourth year of life [12]. An accelerated rate of growth occurs during the first 12 months of fish life, and the juvenile fish can reach the weight of one kilogram by the age of 1 year [13]. Both the nature and quantity of food consumed by *C. idella* are age dependent. During the fry stage (3–5 cm total length, TL), *C. idella* feeds on both zoo- and phytoplankton. However, by the fingerling stage (6–22 cm TL), they entirely switch to plant-based diets [14]. The amount of food ingested by *C. idella* is increased substantially within a few years between the fry (2–3 months old; 0.1 times their body weight) and adult (36–48 months old; up to three times their body weight) stages [15]. The massive digestive activity of adult *C. idella* established them as an organism of choice for controlling unnecessary levels of aquatic plants [16]. Given the crucial role of the buccal cavity in food prehension, figuring out morphologic changes involving its components throughout different stages of fish life will aid in improving their health and productivity. In light of this, the present study aims to define surface ultrastructural and morphometric changes involving the lips, jaws, oral valve, palate, and oral floor as the main components of the buccal cavity. For this purpose, grass carp of three ages, corresponding to fingerling, yearling, and adult stages, were analyzed using scanning electron and light microscopic and morphometric techniques.

## 2. Materials and Methods

### 2.1. Fish Sampling

The present study was carried out on 45 *C. idella* (Valenciennes, 1844) of three different ages: 3 months posthatching (mph; 70 ± 5.6 mm total length, TL), 12 mph (210 ± 10.5 mm TL), and 48 mph (425 ± 21.8 mm TL). These ages correspond to the fingerling, yearling, and adult stages, respectively [13]. The fish were collected from a local farm (Al-Abasa, El-Sharkia governorate, Egypt) during late summer and early autumn of 2022. They were shipped in air-filled plastic containers to the Anatomy Laboratory at the Faculty of Veterinary Medicine at Mansoura University within three hours of their collection. The fish were acclimatized to the laboratory conditions for one week prior to sampling by maintaining them in dimly lit tanks (25–27 °C) with continuous supply of air and food. Fish were humanely killed via immersing them in a chilled water tank containing a lethal dose (250 mg/L) of buffered tricaine methanesulfonate (MS222, E10521, Sigma-Aldrich, St. Louis, MO, USA), according to Wilson et al. [17]. Absence of opercular and fin movements were used as signs for death confirmation.

This study was approved by the Research Ethics Committee of the Faculty of Veterinary Medicine at Mansoura University, Egypt (VM.R. 24.09.177). The study protocols were executed according to the Guide of the National Research Council for the Care and Use of Aquatic Species [18]. Animal experiments were performed in compliance with the ARRIVE guidelines and the NIH Guide for Care and Use of Laboratory Animals (NIH Publications No. 8023, revised 1978).

### 2.2. Gross Morphology and Stereomicroscopy

The roof and floor of the buccal cavity of each fish were excised from the head through two bilateral incisions, starting anteriorly at the corners of the mouth and ending posteriorly just in front of the eyes. The excised tissues were freed from debris, washed in phosphate-buffered saline (PBS), and kept in labeled glass containers containing 10% neutral buffer formalin (NBF) for one week. Different anatomical components of the buccal cavity were identified using a stereomicroscope (SZ61 zoom, Olympus, Tokyo, Japan).

### 2.3. Histological and Histochemical Analysis

Following their fixation in 10% NBF, the roof and floor of the buccal cavity were sectioned at the midline, dehydrated in ascending grades of ethyl alcohol, cleared in xylene, and embedded in molten paraffin. Next, 4 µm thick sections were cut using a rotary microtome (RM2125, Leica Microsystems, Wetzlar, Germany), mounted on glass slides, and stained with Harris’ hematoxylin and eosin (for assessment of general tissue architecture) [19], periodic acid–Schiff (PAS, for evaluation of neutral mucopolysaccharides) [20], and alcian blue pH 2.5 (for evaluation of acidic mucopolysaccharides) [21], as described previously [22]. The stained sections were examined and photographed using a light microscope (Leica DM3000, Leica Microsystems).

### 2.4. Scanning Electron Microscopy Analysis

To elucidate age-specific surface ultrastructural features of the buccal cavity in *C. idella*, various anatomical components of the buccal cavity including the lips, jaws, palate, floor, and tongue were analyzed using scanning electron microscopy, as previously described [4,23]. Briefly, tissue specimens were trimmed and fixed in a solution containing 2.5% paraformaldehyde and 2.5% glutaraldehyde (pH 7.3) at 4 °C. Next, the specimens were re-fixed in 1% osmium tetroxide and dehydrated in acetone and isoamyl acetate. Dehydrated samples were then dried using the critical point drying technique before finally being coated with gold particles. The coated samples were analyzed and photographed using a JSM-6510 LV scanning electron microscope (JEOL Ltd., Tokyo, Japan) located at the Electron Microscopy Unit of Mansoura University. For better contrast, scanning electron micrographs were colored using the quick selection and color fill tools of Adobe Photoshop CS6 software (Adobe Systems, Inc., San Jose, CA, USA), as shown previously [24].

### 2.5. Morphometric Analysis

Different parts of the buccal cavity of *C. idella* were determined in ×10 and ×40 light microscopic images of H&E-, PAS-, and alcian blue-stained sections. The following criteria helped to identify the extent of each of these parts: the upper lip extended between the nares and the upper jaw covered with the common integument externally and lined with buccal mucus membrane internally; the upper jaw was lined by keratinized epithelium and intervened between the upper lip and oral valve; the oral valve appeared as a crescent-shaped projection hanging from the roof of the buccal cavity; the palate formed the remaining part of the buccal roof; the lower lip was represented by the part located rostral to the keratinized lower jaw; and the oral floor was denoted by the expansive part located caudal to the lower jaw till the gill base. The morphometric parameters evaluated by the present study included the thickness of the buccal epithelium and the number of taste buds and goblet cells per every 1 mm^2^ of the buccal epithelium. Micrometric measurements and cell quantifications were calculated using the freehand selection and multipoint counting tools of the ImageJ2 software (ver. 2.14/1.54f) [4,25,26].

### 2.6. Statistical Analysis

Data obtained by the present study were analyzed using the GraphPad Prism 8 (GraphPad Software, La Jolla, CA, USA). Differences between fingerling, yearling, and adult fish groups were detected using one-way ANOVA followed by Tukey’s multiple comparison test. Results were presented as means ± SD, and *p* values less than 0.05 were used to indicate statistical significance.

## 3. Results

### 3.1. Gross Appearance of the Mouth in Fingerling, Yearling, and Adult C. idella

The mouth of *C. idella* appeared to have a terminal position in the three studied ages (Figure 1). Notably, the pigmentation of the lips revealed an age-associated increase. Compared to the fingerling and yearling *C. idella,* which tended to have light gray-colored lips, the lips of adult *C. idella* showed a dark gray coloration (Figure 1).

### 3.2. Surface Ultrastructure of the Buccal Cavity in Fingerling, Yearling, and Adult C. idella

Scanning electron microscopy analysis of components forming the buccal cavity of *C. idella* revealed distinctive surface ultrastructure features for each component (Figure 2, Figure 3, Figure 4, Figure 5, Figure 6 and Figure 7). The roof of the buccal cavity was formed by the upper lip, the upper jaw, the oral valve, and the rostral palate. Meanwhile, the floor of the buccal cavity consisted of the lower lip, the lower jaw, the oral floor, and the tongue.

#### 3.2.1. *C. idella* Fingerlings

##### Roof of the Buccal Cavity

In the three-month-old *C. idella* juveniles, the upper lip, jaw, and oral valve were clearly identified (Figure 2A,B). Type I taste buds (tbIs), a type of taste buds surrounded by a well-developed papilla raised above the level of epithelium bearing taste receptors at its apex, were observed in the mucosa of the upper lip and oral valve, but not in that of the jaw (Figure 2C–E). The superficial epithelial cells of the latter appeared compact, featuring ridge-like borders, and occasionally presented mucus aggregates (Figure 2D). The rostral palatine mucosa formed longitudinal folds separated from each other by wide and shallow grooves (Figure 2F). The folds positioned toward the midline of the palate were curved medially (Figure 2F). Both tbIs and type II taste buds (tbIIs), a type of taste bud in which taste receptors are located on less prominent papillae than tbIs, were detected within the palatine mucosa (Figure 2G,H).

##### Floor of the Buccal Cavity

The lower lip and jaw of three-month-old *C. idella* formed a large ridge-like structure bounding the rostral part of the buccal cavity. A median bulge was extended caudally from the jaw toward the oral floor (Figure 3A). The mucosa of the oral floor was arranged into transverse folds that revealed concave caudal borders (Figure 3A). Several mucus-filled unicellular glands were recognized among the keratinocytes of the lower lip (Figure 3B). Type I taste buds (tbIs) were observed within the mucosa of the lower lip and along the mucosal folds of the oral floor (Figure 3C–F). These buds were surrounded by several openings of goblet cells, especially on the lips (Figure 3D). The mucosa of the lower jaw appeared devoid of taste buds (Figure 3E). The tongue had a bullet shape, filled the median part of the oral floor, and was bordered by longitudinal mucosal folds that converged around its apex (Figure 3G,H). The lingual mucosa appeared corrugated and carried ill-defined longitudinal folds, especially at the rostral end of the tongue (Figure 3G). The mucosa of the tongue was studded with tbIs that appeared to have a decreasing count toward the caudal part of the tongue (Figure 3I–N). Regarding the lingual keratinocytes, they featured tightly packed surface microridges and were separated from each other by distinct ridge-like borders (Figure 3O). Conical papillae were occasionally detected on the tongue, especially toward its caudal part (Figure 3P).

#### 3.2.2. *C. idella* Yearlings

##### Roof of the Buccal Cavity

In the 12-month-old *C. idella* juveniles, the outer surface of the upper lip revealed the presence of numerous type I taste buds (tbIs) surrounded by irregularly outlined papillae that gave them a tulip-like appearance (Figure 4A,B). The epithelium of the outer lip seemed to have high secretory activity, as its surface presented a large number of microtubercles (Figure 4B). The inner surface of the upper lip showed tbIs of a differing appearance from that of the outer side of the lip, where the papillae appeared more compact and had regular outlines (Figure 4C). Compared to the fingerling stage, the upper jaw cells of the yearling *C. idella* revealed progressive changes in their appearance, and their surface was marked with several microdepressions (Figure 4D,E). The jaw cells were separated from each other by deep and continuous microgrooves (Figure 4E). The oral valve showed a large number of tbIs and numerous goblet cells (Figure 4F,G). The longitudinal palatine folds appeared broader than those seen in the fingerling stage and were separated from each other by apparently tighter grooves of corrugated floors (Figure 4H). The mucosa of palatine folds, but not the grooves, were mounted by several tbIs (Figure 4I).

##### Floor of the Buccal Cavity

Stereomicroscopically, the floor of the buccal cavity of *C. idella* yearlings revealed dotted pigmentations (Figure 5A). The jaw was marked by visible lines that appeared perpendicular to the jaw axis (Figure 5A). Transition from the lip into the jaw and from the jaw to the oral floor was clearly evident by this age (Figure 5A,B). Type I taste buds (tbIs) were detected in the mucosa of the lower lip (Figure 5C). Longitudinal electron-dense striations representing the distinctions between jaw ridges were detected (Figure 5D). The jaw cells showed numerous surface microdepressions, and their intercellular boundaries appeared prominently grooved (Figure 5E). The transverse folds of the rostral oral floor were separated from each other by deep and narrow grooves, and their mucosa contained tbIs (Figure 5F,G). The tongue appeared more defined and had a higher position within the oral floor (Figure 5H). Its surface ridges appeared finer and more widely spaced compared to those of the fingerling stage (Figure 5H). The lingual mucosa contained tbIs, a number of which were partially surrounded by several goblet cell openings (Figure 5I,K,L). Regarding the lingual keratinocytes, their surface formed a honeycomb-like structure and appeared bordered from each other by ridge-like structures (Figure 5J). The surface of each keratinocyte revealed several microridges arranged into a fingerprint-like pattern (Figure 5J).

#### 3.2.3. Adult *C. idella*

##### Roof of the Buccal Cavity

Scanning electron microscopy analysis of the upper lip mucosa revealed the presence of several type I taste buds (tbIs) (Figure 6A,B). The upper jaw mucosa was undulated and formed dome-shaped cellular masses separated from each other by well-defined furrows (Figure 6C). The superficial keratinocytes of these jaw masses were separated from each other by more profound microgutters than those seen in the yearling stage (Figure 6D). The surface epithelium of the oral valve and palate appeared more compact than the previous ages and contained several tbIs (Figure 6E–H).

##### Floor of the Buccal Cavity

Grossly, the lower jaw of the adult *C. idella* appeared flattened and firmer in consistency than the other components of the oral floor (Figure 7A). Similar to that seen in the upper jaw, the lower jaw mucosa appeared organized into dome-shaped cellular masses limited from each other by well-defined furrows (Figure 7B). The keratinocytes of the superficial jaw epithelium revealed corrugated surfaces due to the presence of numerous microdepressions (Figure 7C). The folded mucosa of the oral floor and tongue revealed the presence of type I taste buds (tbIs) (Figure 7D–G). Conical papillae and mucus aggregates were sporadically observed on the dorsum linguae (Figure 7G). The lingual keratinocytes carried surface microridges arranged in a fingerprint-like pattern and were separated from each other by clear and thick ridge-like borders (Figure 7H).

A summary of the main surface ultrastructural characteristics of the buccal cavity of *C. idella* during the fingerling, yearling, and adult stages is shown in Table 1.

### 3.3. Histological Features of the Buccal Cavity in Fingerling, Yearling, and Adult C. idella

Histological and histochemical analysis of the buccal cavity of *C. idella* confirmed several important features related to organization of the mucosa and distribution of goblet cells and taste buds (Figure 8, Figure 9 and Figure 10).

#### 3.3.1. *C. idella* Fingerlings

The mucosa of the oral cavity of the three-month-old *C. idella* juveniles consisted of a stratified squamous epithelium that rested on a clear basement membrane overlying a dense connective tissue lamina propria (Figure 8A–C,G,H). Analysis of longitudinal sections of the buccal cavity revealed the presence of bony support to the core of both the upper and lower jaws by means of the premaxilla and the dentary bone, respectively (Figure 8A,G,J). Type I taste buds (tbIs) and goblet cells that expressed both neutral and acidic mucins were within the epithelia of the upper and lower lips (Figure 8B–D,H,I). The epithelial linings of the upper and lower jaws appeared slightly keratinized, and their basal surfaces received a number of dermal papillae from the underlying lamina propria (Figure 8E,J). The oral valve lacked any bony support (Figure 8A). The valve epithelium was remarkably thicker along its labial surface than its palatal one. The epithelium of the latter surface was enriched with goblet cells, while that of the former surface contained taste buds at a higher frequency (Figure 8F). The mucosa of the oral floor appeared folded, and its epithelium was studded with goblet cells of both types (Figure 8K,L).

#### 3.3.2. *C. idella* Yearlings

In 12-month-old *C. idella* juveniles, the epithelium of the upper lip contained both type I taste buds (tbIs) and goblet cells (Figure 9A). The latter expressed neutral mucins and were concentrated within the apical parts of the epithelium (Figure 9B). The upper jaw epithelium appeared moderately keratinized (Figure 9C). The epithelium of the oral valve was non-keratinized and was clearly demarcated from the surrounding components of the oral roof (Figure 9C–E). The goblet cells were numerous and displaced toward the superficial layer of the epithelium of the palatine side of the valve (Figure 9E,F). The lamina epithelialis of the lower lip contained tbIs (Figure 9G,H). Chromatophores were seen at the junction of the epithelium with the superficial layer of the lamina propria (Figure 9H). The keratinized epithelium of the lower jaw was invaginated by several papillae from the underlying connective tissue layer (Figure 9I). The tongue mucosa was folded and studded with several goblet cells of both acidic and neutral types (Figure 9J–L).

#### 3.3.3. Adult *C. idella*

In the 48-month-old *C. idella*, the epithelium of the upper lip contained type I taste buds (tbIs) and goblet cells with prominent secretory activity (Figure 10A,B). The upper jaw epithelium was heavily keratinized, and its superficial layer showed several cornified cells separated from each other by well-defined microgrooves (Figure 10C). The oral valve contained goblet cells and taste buds with the same preferential locations for each of them seen at the yearling stage (Figure 10D,E). However, both goblet cells and taste buds seemed to fade out toward the free end of the valve (Figure 10F). The goblet cells of the palatal epithelium were increased at the basal regions of mucosal grooves (Figure 10G). The epithelium of the lower jaw was organized into dome-shaped masses separated from each other by deep furrows. Moreover, its basal layer was folded and received a number of dermal papillae from the underlying lamina propria (Figure 10H). The cornified cells of the superficial epithelium were demarcated from each other by well-defined microgrooves of almost similar depth to those of the upper jaw (Figure 10I). The lamina propria of the tongue revealed a remarkable increase in thickness and separated the epithelium from the tongue musculature (Figure 10J). TbIs were observed in the mucosa of the tongue (Figure 10K). The intrinsic muscles of the tongue consisted of transverse muscle fibers rostrally and longitudinal muscle fibers caudally (Figure 10L).

### 3.4. Morphometry of the Buccal Mucosa in Fingerling, Yearling, and Adult C. idella

The thickness of the epithelium of the buccal cavity of *C. idella* revealed both spatial and temporal changes (Figure 11). In fingerlings, yearlings, and adults, the thickest epithelium was found in the jaws, followed by the lips, while the palate and the palatal side of the oral valve were the thinnest. Although the epithelial thickness of different components of the buccal cavity significantly increased from the fingerling stage to the yearling stage and from the latter to the adult stage, the changes involving the epithelial thickness of the upper and lower jaws were the most noticeable (Figure 11).

The number of taste buds was remarkably higher within the palatal and oral valve epithelia, specifically along the palatine surface of the valve, compared to those of the upper and lower lips. No significant changes were observed in between stages (Figure 12).

The numbers of the acidic and neutral goblet cells were almost equal within the epithelium of each component of the buccal cavity (Table 2). The number of goblet cells in the labial, palatal, and floor epithelia did not significantly differ between the fingerling and the yearling stages. However, they increased significantly from the latter to the adult stage. Conversely, the number of goblet cells in the oral valve epithelium significantly increased between the three studied ages (Table 2).

## 4. Discussion

The buccal cavity of fish plays an important role in food perception and fish adaptation to various environmental conditions [27]. Surface ultrastructural features of the buccal cavity have been described by various studies in teleost with differing feeding habits that include those of Alsafy et al. [28] and Madkour et al. [29] in carnivorous species; El Bakary [30], Yashpal et al. [31], and Elsheikh et al. [32] in herbivorous species; and Sayed et al. [33] and Gamal et al. [5] in omnivorous species. This study shows significant modifications in the surface architecture at different regions of the buccal cavity of the herbivorous cyprinid *C. idella* and the effect of age on its various components using light, stereomicroscopic, and scanning electron microscopic techniques.

The lips of vertebrates play essential roles in capturing food particles before passing them to the lumen of the buccal cavity. The structure and morphological appearance of lips vary greatly among species depending on the mouth type and dietary behavior of the fish [34]. Using SEM analysis, the present study revealed the upper and lower lips to consist of continuous structures bounding the most anterior part of the buccal cavity without any evidence of any interruption at the midline of each lip. This observation corresponds with those of Tripathi and Mittal [34] in common carp (*Cyprinus carpio*), Yashpal et al. [31] in mrigal carp (*Cirrhinus mrigala*), and Gamal et al. [5] in African catfish (*Clarias gariepinus*). However, our finding is different from those reported in several teleost species, mainly of carnivorous behavior, which reported the presence of a median fissure dividing the upper lip into two halves in Rita catfish (*Rita rita*) [35], white grouper (*Epinephelus aeneus*) [36], marbled spinefoot (*Siganus rivulatus*) [33], and grey gurnard (*Eutrigla gurnardus*) [37]. Such differences among species could represent adaptations of fish to their dietary habits. Moreover, our SEM analysis of *C. idella* lips confirmed their secretory activity as early as the fingerling and yearling stages, where epithelium appeared studied with several goblet cells, mucus particles, and surface microtubercles.

The jaw epithelium of *C. idella* was the thickest among all other components of the buccal cavity during fingerling, yearling, and adult stages. Moreover, its keratinized superficial layer revealed an age-associated increase in its thickness. Starting from the yearling stage, it appeared to be separated by depressed intercellular boundaries. These boundaries were only evident ultrastructurally in yearling fish but both histologically and ultrastructurally in adult ones. The noted interdigitations between the undulating basal region of the jaw epithelium and the underlying papillary dermis could provide a sort of fixation to the epithelium during food grasping and biting. This interdigitating pattern appeared more complex with advancing fish age and corresponds with the well-known pattern involving dermal–epidermal junctions in human skin [38].

The oral valve or velum of *C. idella* appeared as a single crescent-shaped mucosal flap of almost equal length derived from the roof of the buccal cavity along the posterior edge of the upper jaw. Consistent with this study, the presence of an oral valve connected to the upper jaw has been noted in mrigal carp (*C. mrigala*) [31]. However, the presence of an additional valve along the posterior aspect of the lower jaw has been noted in the buccal cavity of the bayad (*Bagrus bayad*) [39], the mola carplet (*Amblypharyngodon mola*) [40], the grey gurnard (*Eutrigla gurnardus*) [37], the white grouper (*E. aeneus*) [28], and the largehead hairtail (*Trichiurus lepturus*) [41]. Furthermore, the upper velum of the latter mentioned species appeared to be divided into rostral and caudal parts [41]. The valvar mucosa lacked any muscular or skeletal support in *C. idella,* as revealed by the present study. Taken together, both morphological and structural differences exist in the oral valve of fish of different dietary behaviors. These differences should be considered during the assessment of various pathologies in the buccal cavity of fish.

The appearance and degree of development of the tongue vary among fish species. The tongue of *C. idella* was found to undergo morphological changes throughout posthatching life. It appeared as a bullet-shaped structure filling the median part of the oral floor and surrounded by the longitudinal mucosal folds of the oral floor. Its mucosa appeared folded and seeded with a large number of taste buds along the dorsal surface of these folds, indicating a gustatory role for the tongue. Such position of taste buds within the tongue of *C. idella* could help to ensure maximum contact between these sensory structures and food particles upon encountering them, thereby deciding their fate with the digestive tract, whether to pass them to the pharynx or to return them to the outside of the body. By the yearling age, it appeared more defined, with extended outlines and an almost flat surface. Its bulk is formed by a number of skeletal muscle bundles. The epithelial cells of the tongue carried surface microridges and appeared separated from each other by distinct ridges. Similar findings were observed in the tongue of turbot (*Scophthalmus maximus*) [42]. The dorsal surface of the tongue appeared minimally papillated throughout the fingerling stage to the adult stage. No lingual teeth were detected. Different from the present study, the muscular tongue of the carnivorous fish European seabass (*Dicentrarchus labrax*) revealed the presence of numerous cone-shaped teeth along its dorsal surface that probably aid in the breakdown of ingested prey [43].

Goblet cells are exocrine unicellular glands that actively discharge membrane-bound mucoid substances. The confinement of goblet cells to the most superficial layers of the buccal epithelium is important for the rapid distribution of their secretions along the mucosal surface [44]. Continuously discharged mucus globules from goblet cell openings help to clean the buccal cavity, lubricate food particles, and facilitate their passage toward the subsequent parts of the digestive tract. Noteworthily, mucus collected from the upper digestive tract of tilapia revealed a similar function to that of mammalian saliva [45]. Other important functions of oral mucus of teleost include solubilization of food, facilitation of its mastication, initial digestion of starches and lipids, and antimicrobial activity [46]. Goblet cells expressing neutral and acidic mucins were noted within the buccal mucosa, being highest in the upper lip and palate. The number of both types of goblet cells was almost unchanged between the fingerling and yearling stages; however, they appeared significantly higher in adult fish. As an antipathogenic effect has been suggested for acidic mucus due to the presence of sialic acid residues [47,48], the higher presence of acidic goblet cells may explain the higher resistance of adult fish to pathogenic diseases than juvenile ones [49].

The superficial epithelial cells of the teleost buccal cavity represent the most active layer of the epithelium and are characterized by their ability to secrete surface secretory vesicles [1]. Compared to the cells of the deep layers of the epithelium that contain only tonofilaments, the cells of the superficial layer of the buccal epithelium contain both tonofilaments and actin filaments. The latter are responsible for the formation of microridges or micropapillae at the cell surface [50]. The function of these microridges is unknown; however, they may protect against mechanical trauma and help in mucous retention to the fish surface [51]. In view of this, surface microridges arranged in a fingerprint-like pattern on the keratinocytes of the buccal epithelium were observed during different stages of posthatching life in *C. idella* [39].

Taste buds are formed by the aggregation of a number of receptor cells. Three types of taste buds (I, II, and III) have been described in the buccopharyngeal cavity of teleost [52]. In Nile tilapia (*Oreochromis niloticus*), the type I taste buds are located on high epidermal papillae and distributed within the mucosa of the rostral part of the buccal cavity [32]. The other two types are seldom seen above the epithelium, possess low epidermal papillae, and are localized to the mucosa of the metabranchial region of the buccopharyngeal cavity, part of the cavity just caudal to the gill box. Type I and II taste buds were suggested to act as chemoreceptors and mechanoreceptors, while type III taste buds are predominantly chemoreceptors. In this regard, the taste buds, mainly of type I, were observed in the mucosa of different components of the buccal cavity. Our morphometric evaluation of their number revealed the highest abundance of taste buds in the mucosa of the palate and oral floor. This finding indicates that these portions of the buccal cavity evolved a peculiar gustatory activity that helps with the preferential selection of food. A similar distribution of goblet cells was seen in the mola carplet (*A. mola*) [40].

## 5. Conclusions

In summary, this study reported for the first time several age-associated surface ultrastructural, histological, and morphometric changes in the buccal cavity components of *C. idella*. These changes could be due to exposure to massive amounts of food from the fingerling stage to the adult stage. The present study improves knowledge about the functional anatomy of the oral cavity in cyprinids and will serve as a reference for evaluating mechanisms of food prehension in them.

## Figures and Tables

**Figure 1 animals-14-03162-f001:**
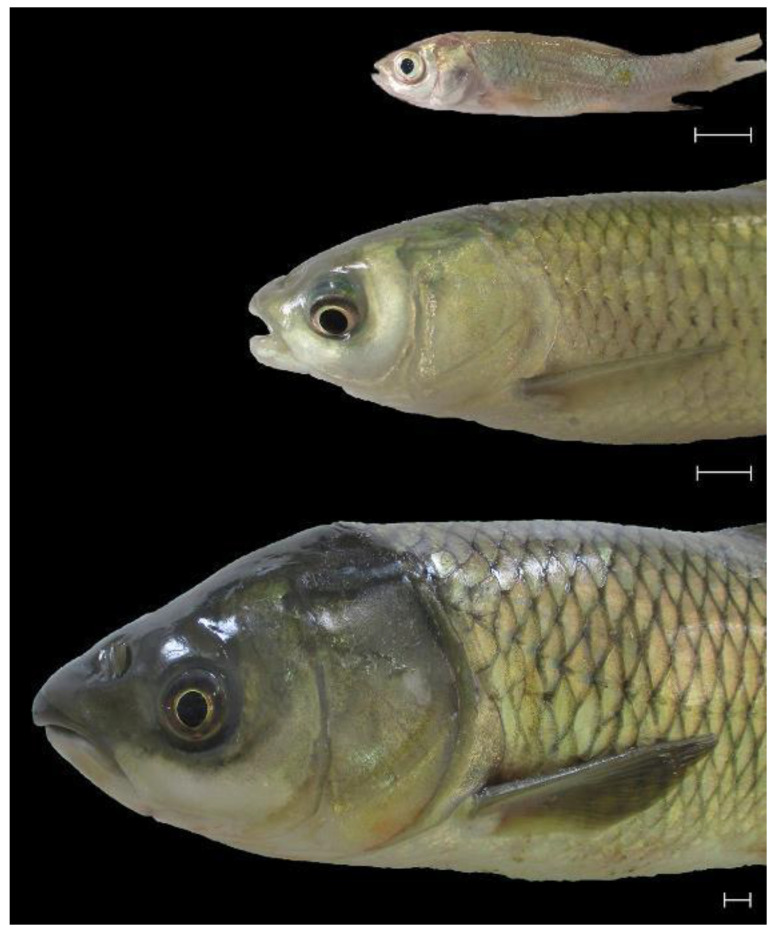
The three ages of *C. idella* used in the present study: 3 months (fingerlings; top), 12 months (yearlings; middle), and 48 months (adults; bottom). Note the terminal position of the mouth. Scale bar = 10 mm.

**Figure 2 animals-14-03162-f002:**
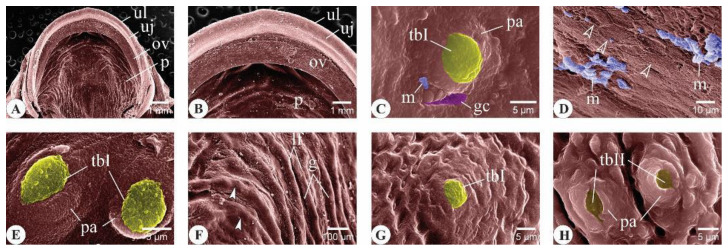
Surface ultrastructural features of the upper lip, upper jaw, oral valve, and rostral palate in three-month-old *C. idella* juveniles (fingerlings). (**A**) Scanning electron micrograph (SEM) of the upper lip (ul), upper jaw (uj), oral valve (ov), and rostral palate (p). (**B**) Higher magnification of (**A**). (**C**) SEM of the upper lip mucosa. (**D**) SEM of the superficial epithelial cells of the upper jaw showing the distinct ridge-like borders (empty arrowheads). (**E**) SEM of the oral valve mucosa. (**F**) SEM of the palate mucosa showing longitudinal folds (lf) intervened by wide and shallow grooves (g). The curved ends of the folds located toward the median part of the palate are denoted by arrowheads. (**G**,**H**) SEMs of the palate mucosa showing taste buds of types I (**G**) and II (**H**). gc, goblet cell opening; m, mucus; pa, papilla; tbI, type I taste bud; tbII, type II taste bud.

**Figure 3 animals-14-03162-f003:**
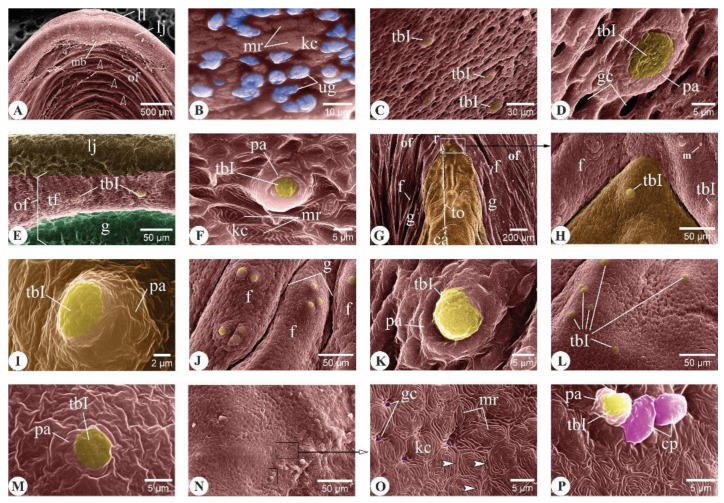
Surface ultrastructural features of the lower lip, lower jaw, oral floor, and tongue in three-month-old *C. idella* juveniles (fingerlings). (**A**) Scanning electron micrograph (SEM) of the lower lip (ll), lower jaw (lj), and oral floor (of). The mucosa of the oral floor is arranged into transverse folds (empty arrowheads). (**B**) SEM of the lower lip mucosa showing numerous mucus-filled unicellular glands (ug). (**C**) SEM of the lower lip mucosa. (**D**) High magnification of (**C**). (**E**) SEM of the lower jaw (lj) and oral floor (of) mucosa. (**F**) High magnification of (**E**). (**G**) SEM of the tongue (to) and oral floor (of). r, rostral; ca, caudal. (**H**) SEM of the rostral tongue mucosa. (**I**) High magnification of (**H**). (**J**) SEM of the latero-rostral tongue mucosa. (**K**) High magnification of (**J**). (**L**) SEM of the mid-dorsal tongue mucosa. (**M**) High magnification of (**L**). (**N**) SEM of the caudal tongue mucosa. (**O**) High magnification of (**N**). Note the distinct ridge-like borders of lingual keratinocytes (arrowheads). (**P**) High magnification of (**N**) (low boxed area). f, fold; g, groove; gc, goblet cell opening; kc, keratinocytes; m, mucus; mb, median bulge; mr, microridges; pa, papilla; tf, transverse folds; tbI, type I taste bud.

**Figure 4 animals-14-03162-f004:**
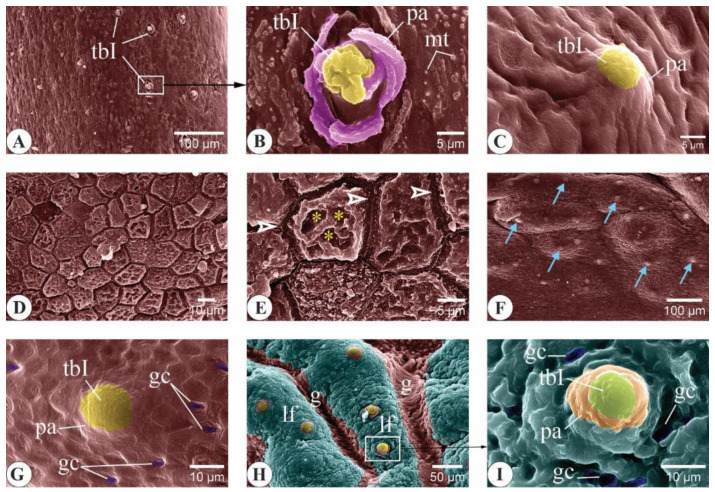
Surface ultrastructural features of the upper lip, upper jaw, oral valve, and palate in 12-month-old *C. idella* juveniles (yearlings). (**A**) Scanning electron micrograph (SEM) of the outer surface of the upper lip. (**B**) High magnification of (**A**). (**C**) SEM of the inner surface of the upper lip. (**D**) SEM of the upper jaw mucosa. (**E**) High magnification of (**D**). Note the surface microdepressions (*) of jaw cells and the continuous microgutter-like intercellular grooves (empty arrowheads). (**F**) SEM of the oral valve showing numerous taste buds (blue arrows). (**G**) High magnification of (**F**). (**H**) SEM showing the palate to consist of longitudinal folds (lf) separated from each other by deep and narrow grooves (g). (**I**) High magnification of (**H**). gc, goblet cell opening; mt, microtubercles; pa, papilla; tbI, type I taste bud.

**Figure 5 animals-14-03162-f005:**
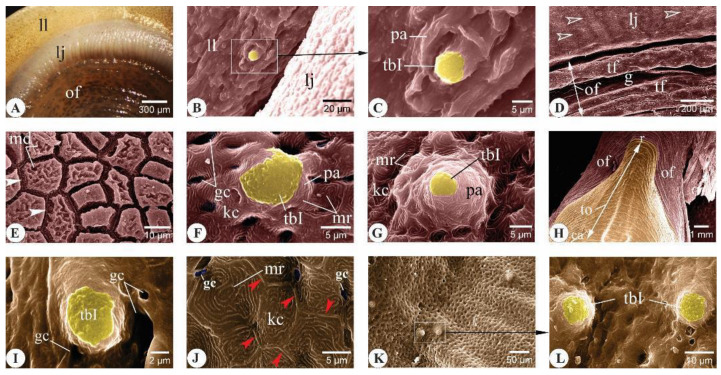
Surface ultrastructural features of the lower lip, lower jaw, oral floor, and tongue in 12-month-old *C. idella* juveniles (yearlings). (**A**) A stereomicrograph of the lower lip (ll), lower jaw (lj), and oral floor (of). (**B**) Scanning electron micrograph (SEM) of the lower lip (ll) and lower jaw (lj) mucosa. (**C**) High magnification of (**B**). (**D**) SEM of the lower jaw (lj) and oral floor (of). The jaw mucosa is marked by longitudinal electron-dense striations (empty arrowheads). (**E**) SEM of the lower jaw mucosa showing surface microdepressions (md) and deep and continuous gutter-like intercellular boundaries (arrowheads). (**F**,**G**) SEMs of the mucosa of the rostral (**F**) and lateral (**G**) parts of the oral floor. (**H**) SEM of the tongue (to) and oral floor (of). r, rostral; ca, caudal. (**I**) SEM of the rostrolateral tongue mucosa. (**J**) SEM of the mid-dorsal tongue mucosa showing the honeycomb-like structures formed by lingual keratinocytes (kc). The latter are bordered by ridge-like structures (red arrowheads) and carry surface microridges (mr) arranged into a fingerprint-like pattern. (**K**) SEM of the latero-caudal tongue mucosa. (**L**) High magnification of (**K**). g, groove; gc, goblet cell opening; kc, keratinocytes; mr, microridges; pa, papilla; tbI, type I taste bud; tf, transverse folds.

**Figure 6 animals-14-03162-f006:**
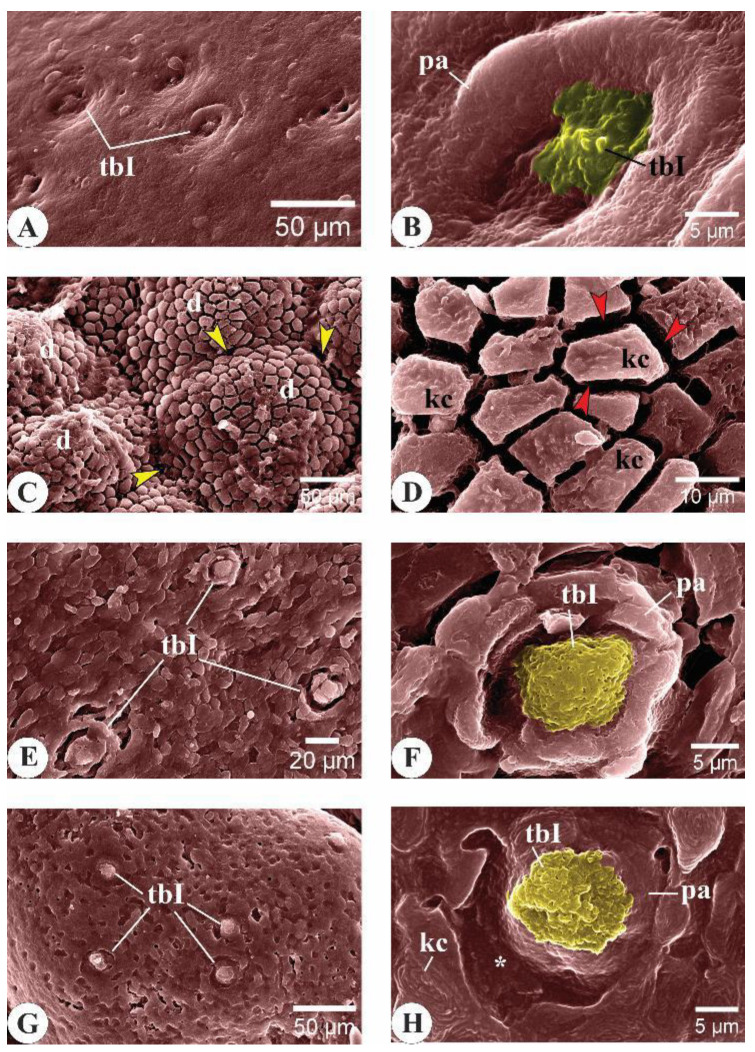
Surface ultrastructural features of the upper lip, upper jaw, oral valve, and rostral palate in 48-month-old *C. idella* (adults). (**A**,**B**) Scanning electron micrographs (SEMs) of the upper lip. (**C**) SEM of the upper jaw showing dome-shaped cellular masses (d) separated from each other by well-defined furrows (yellow arrowheads). (**D**) High magnification of (**C**) showing the superficial keratinocytes (kc) of the jaw are bounded by deep and continuous microgutters (red arrowheads). (**E**,**F**) SEMs of the oral valve mucosa. (**G**,**H**) SEMs of the palate mucosa showing a depressed area preceding a taste bud (*). kc, keratinocytes; pa, papilla; tbI, type I taste bud.

**Figure 7 animals-14-03162-f007:**
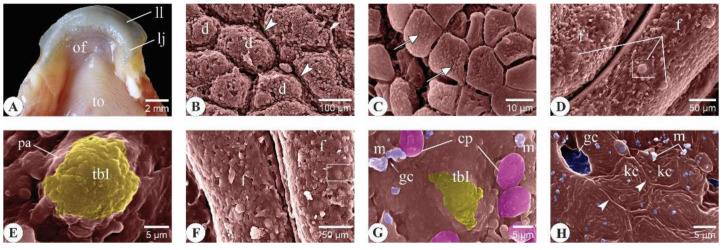
Surface ultrastructural features of the lower lip, oral floor, and tongue in 48-month-old *C. idella* (adults). (**A**) A photograph of the lower lip (ll), oral jaw (lj), oral floor (of), and tongue (to). (**B**) SEM of the lower jaw showing dome-shaped cellular masses (d) surrounded by well-defined furrows (arrowheads). (**C**) High magnification of (**B**) showing the corrugated surface of the superficial jaw epithelium (arrows). (**D**) SEM of the oral floor mucosa. (**E**) High magnification of the boxed area in (**D**). (**F**) SEM of the tongue mucosa. (**G**) High magnification of the boxed area in (**F**). (**H**) SEM of the tongue mucosa showing lingual keratinocytes (kc) bordered by clear ridge-like structures (arrowheads) and carrying surface microridges arranging themselves into a fingerprint-like pattern. cp, conical papillae; f, mucosal fold; gc, goblet cell opening; m, mucus; pa, papilla; tbI, type I taste bud.

**Figure 8 animals-14-03162-f008:**
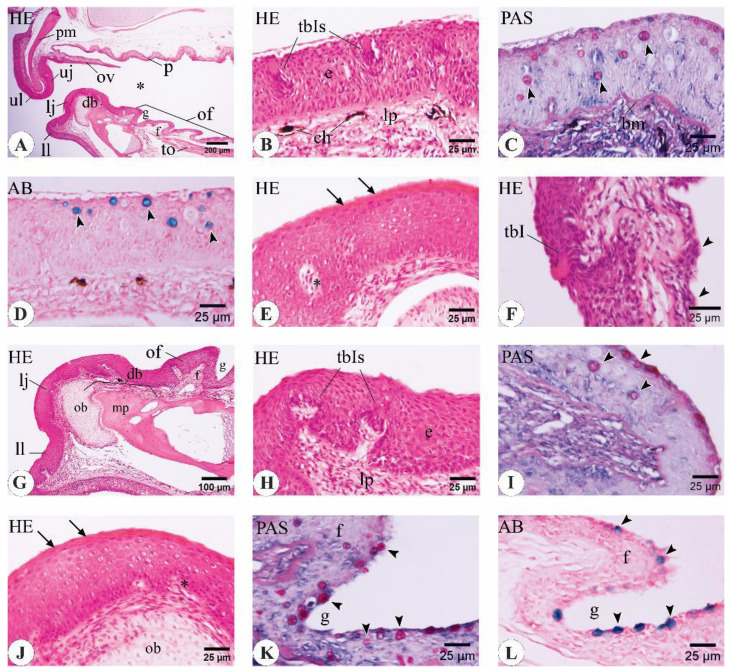
Important histological and histochemical features of the buccal cavity in three-month-old *C. idella* juveniles (fingerlings). (**A**) Longitudinal section of the upper lip (ul), upper jaw (uj), oral valve (ov), rostral palate (p), lower lip (ll), lower jaw (lj), oral floor (of), and tongue (to). The lumen of the buccal cavity is indicated by an asterisk. (**B**) The epithelium (e) of the upper lip contains type I taste buds (tbIs). (**C**) The epithelium of the upper lip contains neutral goblet cells (arrowheads) and rests on a distinct basement membrane (bm). (**D**) Acidic goblet cells (arrowheads) within the upper lip epithelium. (**E**) The keratinized epithelium of the upper jaw (arrows). *, dermal papillae. (**F**) Goblet cells (arrowheads) and a type I taste bud (tbI) within the epithelium of the oral valve. (**G**) Longitudinal section of the lower lip (ll), lower jaw (lj), and oral floor (of). (**H**) The epithelium of the lower lip contains tbIs. (**I**) Neutral goblet cells within the epithelium of the lower lip. (**J**) The keratinized epithelium of the lower jaw (arrows). *, dermal papillae. (**K**) Neutral goblet cells within the lower lip epithelium (arrowheads). (**L**) Acidic goblet cells within the lower lip epithelium (arrowheads). ch, chromatophores; db, dentary bone; f, fold; g, groove; lp, lamina propria; mp, mineralized part of db; ob, osteoblasts cluster of db; pm, premaxilla. Stains are hematoxylin and eosin (HE; **A**, **B**, **E**–**H**, **J**), periodic acid–Schiff (PAS; **C**, **I**, **K**), and alcian blue (AB; **D**, **L**).

**Figure 9 animals-14-03162-f009:**
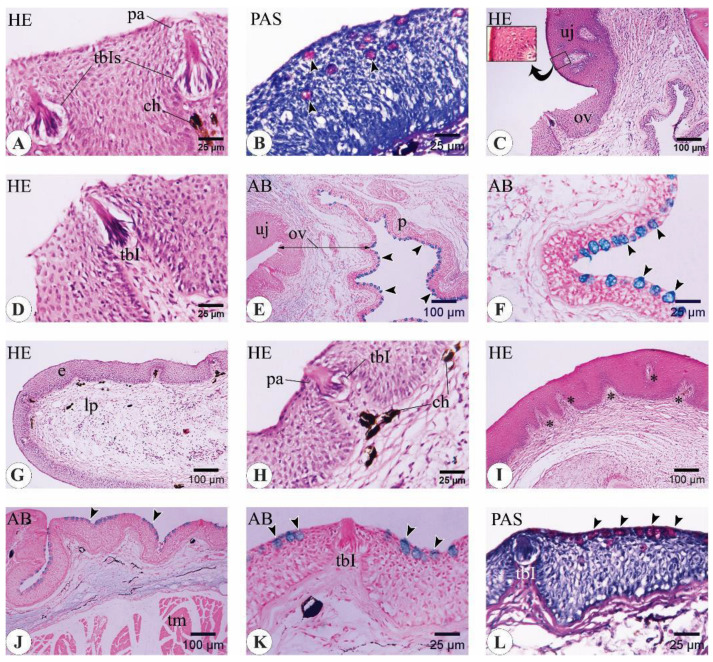
Important histological and histochemical features of the buccal cavity in 12-month-old *C. idella* juveniles (yearlings). (**A**,**B**) The epithelium of the upper lip contains type I taste buds (tbIs) and neutral goblet cells (arrowheads in **B**). (**C**) Longitudinal section in the upper jaw (uj) and oral valve (ov). Note the cornified surface of the jaw epithelium (inset). (**D**) A type I taste bud (tbI) in the epithelium of the oral valve. (**E**) Longitudinal section in the uj, ov, and rostral palate (p). Acidic goblet cells are scarce on the uj and labial surface of the ov but numerous on the palate and the palatine side of the ov (arrowheads). (**F**) Acidic goblet cells are concentrated in the superficial parts of the valve epithelium (arrowheads). (**G**) The mucosa of the lower lip consists of a lamina epithelialis (e) superimposed on a lamina propria (lp). (**H**) The epithelium of the lower lip contains a tbI. (**I**) The keratinized epithelium of the lower jaw is invaginated by several dermal papillae (*). (**J**–**L**) The folded mucosa of the tongue is studded with several goblet cells expressing acidic (**J**,**K**) and neutral (**L**) types (arrowheads) and a variable number of tbIs. ch, chromatophores; tm, tongue musculature. Stains are hematoxylin and eosin (HE; **A**, **C**, **D**, **G**–**I**), periodic acid–Schiff (PAS; **B**, **L**), and alcian blue (AB; **E**, **F**, **J**, **K**).

**Figure 10 animals-14-03162-f010:**
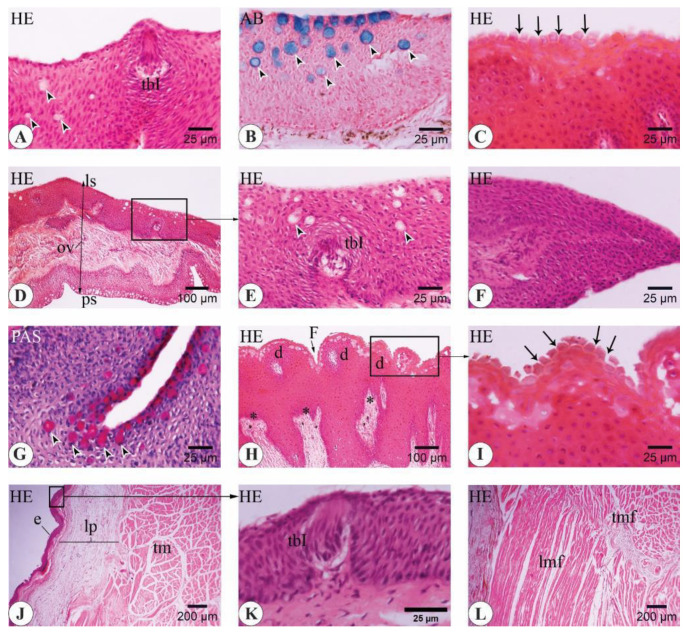
Important histological and histochemical features of the buccal cavity in 48-month-old *C. idella* (adults). (**A**,**B**) The epithelium of the upper lip contains a type I taste bud (tbI) and acidic goblet cells (arrowheads). (**C**) The superficial layer of the upper jaw epithelium is deeply cornified, and the cells are surrounded by well-defined microgutters (arrows). (**D**–**F**) Longitudinal sections in the oral valve (ov). The epithelium of the labial surface (ls) of the ov is interspersed with fewer goblet cells (arrowheads) than that of the palatine surface (ps). The presence of taste buds is limited to the ls of the valve. A tbI in the ov epithelium is shown in E. Both goblet cells and taste buds fade out toward the free edge of the ov (**F**). (**G**) Goblet cell density is increased in the basal regions of the palatal mucosal grooves. (**H**) The epithelium of the lower jaw is arranged into dome-shaped masses (d) separated from each other by deep furrows (f). The basal layer of the epithelium is folded and interdigitates with deeper layers by a number of dermal papillae (*). (**I**) The cornified cells of the superficial epithelium are separated by well-defined microgutters (arrows). (**J**) The lamina propria (lp) of the tongue is thickened and separates the epithelium (e) from the tongue musculature (tm). (**K**) A tbI in the mucosa of the tongue. (**L**) The lingual muscles consist mainly of transverse muscle fibers (tmf) rostrally and longitudinal muscle fibers (lmf) caudally. Stains are hematoxylin and eosin (HE; **A**, **C**–**F**, **H**–**L**), periodic acid–Schiff (PAS; **G**), and alcian blue (AB; **B**).

**Figure 11 animals-14-03162-f011:**
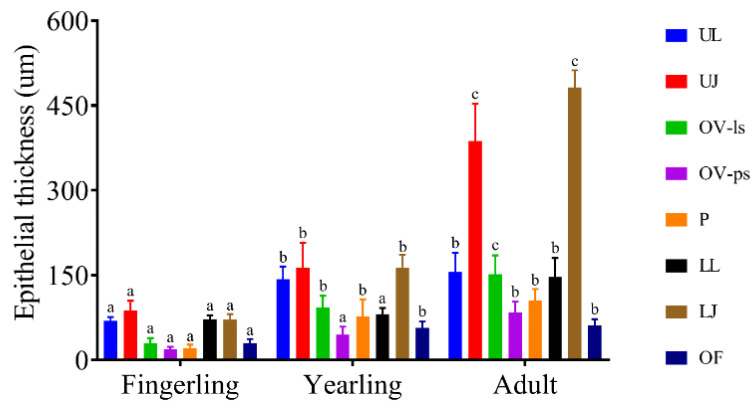
Bar chart showing the changes in the epithelial thickness across different components of the buccal cavity of *C. idella* and in between the fingerling, yearling, and adult stages. UL, upper lip; UJ, upper jaw; OV-ls, labial surface of oral valve; OV-ps, palatal surface of oral valve; P, palate; LL, lower lip; LJ, lower jaw; OF, oral floor. Data are presented as means ± standard deviations. Different superscripts on the same-colored bars indicate statistical significance, *p* < 0.05.

**Figure 12 animals-14-03162-f012:**
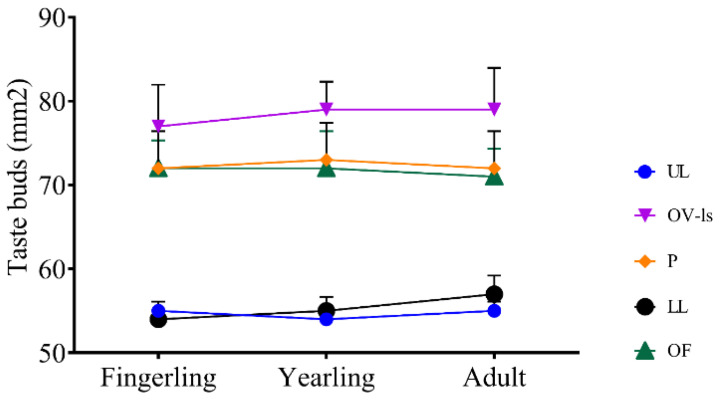
Line graph showing the changes in the taste bud count across different components of the buccal cavity of *C. idella* and in between the fingerling, yearling, and adult stages. UL, upper lip; OV-ls, labial surface of oral valve; P, palate; LL, lower lip; OF, oral floor. The upper and lower jaws and the palatal surface of the oral valve were not included in the count due to the scarcity of taste buds in their epithelia. Data are presented as means ± standard deviations.

**Table 1 animals-14-03162-t001:** Summary of surface ultrastructural features of the buccal cavity of *C. idella* during fingerling, yearling, and adult stages.

Part	Fingerling	Yearling	Adult
UL	- Numerous tbIs and several goblet cell openings were observed in between the epithelial cells.	- As fingerling stage.	- As fingerling stage.
UJ	- Jaw cells carried surface microridges that were mainly clear at the intercellular junctions. - No taste buds or goblet cells were detected.	- Jaw cells revealed several surface microdepressions and were bounded by continuous microgrooves.	- Jaw surface epithelium appeared undulated due to the presence of several dome-shaped cellular masses separated from each other’s by distinct furrows. - Each of these masses was formed of many keratinized cells separated from each other’s by deep microgutters.
V	- Numerous tbIs were observed in between the epithelial cells. - Goblet cell openings were scarce on the labial surface of the valve but abundant on its palatine surface.	- As fingerling stage.	- As fingerling stage.
P	- Mucosa was formed of widely spaced thin longitudinal folds. - Numerous tbIs and occasional tbIIs were detected along the folds.	- Palatine folds appeared broader and more tightly spaced than those of the fingerling stage.	- As yearling stage.
LL	- Several tbIs were detected among the epithelial cells.	- As fingerling stage.	- As fingerling stage.
LJ	- Almost similar to the UJ.	- Almost similar to the UJ.	- Almost similar to the UJ.
OF	- Mucosa was thrown into transverse folds rostral to the tongue and longitudinal folds bilateral to it. - The tongue appeared bullet shaped. - Distribution of tbIs was concentrated along the folds and at the rostral parts of the tongue. - Lingual keratinocytes carried surface microridges arranged in a fingerprint-like pattern.	- The transverse folds were separated from each other’s by deep and narrow grooves. - The tongue appeared more defined with raised outlines and occupied most of the lumen of the buccal cavity.	- As yearling stage.

UL, upper lip; UJ, upper jaw; V, oral valve; P, palate; LL, lower lip; LJ, lower jaw; OF, oral floor; tbIs, type I taste buds; tbIIs, type II taste buds.

**Table 2 animals-14-03162-t002:** Changes in goblet cell count per mm^2^ across different parts of the buccal cavity of *C. idella* during fingerling, yearling, and adult stages.

Part	Type of GC	Fingerling	Yearling	Adult
UL	AGC	526 ± 58 ^a^	583 ± 54 ^a^	913 ± 107 ^b^
	NGC	519 ± 62 ^a^	576 ± 61 ^a^	945 ± 66 ^b^
UJ	AGC	nd	nd	nd
	NGC	nd	nd	nd
OV-ls	AGC	204 ± 26 ^a^	413 ± 95 ^b^	891 ± 61 ^c^
	NGC	201 ± 27 ^a^	399 ± 73 ^b^	884 ± 59 ^c^
OV-ps	AGC	402 ± 44 ^a^	599 ± 61 ^b^	1233 ± 106 ^c^
	NGC	393 ± 58 ^a^	594 ± 47 ^b^	1242 ± 65 ^c^
P	AGC	792 ± 103 ^a^	863 ± 79 ^a^	1624 ± 96 ^b^
	NGC	789 ± 76 ^a^	871 ± 65 ^a^	1631 ± 78 ^b^
LL	AGC	425 ± 101 ^a^	438 ± 66 ^a^	864 ± 65 ^b^
	NGC	417 ± 88 ^a^	451 ± 56 ^a^	858 ± 51 ^b^
LJ	AGC	nd	nd	nd
	NGC	nd	nd	nd
OF	AGC	786 ± 39 ^a^	834 ± 92 ^a^	1318 ± 70 ^b^
	NGC	797 ± 46 ^a^	819 ± 108 ^a^	1323 ± 73 ^b^

GC, goblet cell; AGC, acidic goblet cell; NGC, neutral goblet cell; UL, upper lip; UJ, upper jaw; OV-ls, labial surface of oral valve; OV-ps, palatal surface of oral valve; P, palate; LL, lower lip; LJ, lower jaw; OF, oral floor; nd, not detected. Data are presented as means ± standard deviations. Different superscript letters in the same row indicate statistical significance, *p* < 0.05.

## Data Availability

All data are available from the corresponding author on request.

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
