# Peer review of "Spatiotemporal Ultrastructural, Histological, and Morphometric Changes in the Buccal Cavity of Grass Carp (Ctenopharyngodon idella) During Fingerling, Yearling, and Adult Stages"

_animals, 2024, doi:10.3390/ani14213162_

Round 1
Reviewer 1 Report
Comments and Suggestions for Authors
In my opinion, MS is well planned, described and very interesting. A great advantage of MS and a great value are excellent quality histological and electron microscopic figures. Scanning electron microscopy, histology and morphometry are advanced research methods that allow for very detailed analysis of structural changes. Such studies provide a solid foundation for future research in developmental biology and physiology. The authors described the figures in great detail, which is rarely seen in histological MS. In this study, the authors reported for the first time several age-associated (of fingerling, yearling and adult) surface ultrastructural, histological, and morphometric changes in the buccal cavity components of C. idella. According to the authors, these changes could be due to exposure to massive amounts of food from the fingerling to the adult stage.
Understanding the morphological and functional differences in the oral cavity at different stages of life can help farmers to better adapt feeds and environmental conditions to the needs of different age groups. This in turn can improve the efficiency of fish farming. Knowledge of the presence of goblet cells, which produce mucus, and taste buds can also help to better understand how fish perceive and respond to different types of food, which is crucial in designing feeds for fish farming. Knowledge of the adaptations of the oral cavity of fish can not only improve the efficiency of farming, but also provide valuable clues for diagnosing diseases and pathologies of other fish species. However, before accepting the MS for publication, the authors should improve the MS, in particular check the English language. My minor comments are below:
Line 19: Please delete: or C. Idella
Line 20: Please to replace on “In the present study, different parts of the of the buccal cavity in C. idella were examined using light microscopy and scanning electron microscopy.”
Line 21: Requests to replace "We" on "They have been"
Line 32: Please delete: ”or C. idella”
Line 47: Most of the keywords are repeated with the MS title. Please replace the repeated keywords with e.g.: mouth; vertebrates; digestive tract
Line: 79: Please delete: “or C. idella”Line
Line 112: Please change “aquaria” to “tanks”
Materials and Methods
Line 114: Please add: …..(MS222; E10521……
Line 131: In what percentage of formalin were the samples preserved? Please add this information
Line 131: Were the samples first dehydrated in a series of alcohol, acetone and xylene? Please describe this in more detail
Line 132: How were the samples cut into sections? Using a microtome? Which one? Please provide the information
Discussion
Line 486: Requests to replace “Contrary to what these studies show”
Line 496: Requests to replace: "Consistent observation in this study"
Line 502: Please change “work” to “study”
Line 528: Please change “work” to “study”
Line 552: ’’Our’’ change to ’’ In this study”
Line 562: Change please “work” to “study”
Line 562: Please delete “our”
Line 582: Please change “work” to “study”
Author Response
Comment: In my opinion, MS is well planned, described and very interesting. A great advantage of MS and a great value are excellent quality histological and electron microscopic figures. Scanning electron microscopy, histology and morphometry are advanced research methods that allow for very detailed analysis of structural changes. Such studies provide a solid foundation for future research in developmental biology and physiology. The authors described the figures in great detail, which is rarely seen in histological MS. In this study, the authors reported for the first time several age-associated (of fingerling, yearling and adult) surface ultrastructural, histological, and morphometric changes in the buccal cavity components of C. idella. According to the authors, these changes could be due to exposure to massive amounts of food from the fingerling to the adult stage.
Understanding the morphological and functional differences in the oral cavity at different stages of life can help farmers to better adapt feeds and environmental conditions to the needs of different age groups. This in turn can improve the efficiency of fish farming. Knowledge of the presence of goblet cells, which produce mucus, and taste buds can also help to better understand how fish perceive and respond to different types of food, which is crucial in designing feeds for fish farming. Knowledge of the adaptations of the oral cavity of fish can not only improve the efficiency of farming, but also provide valuable clues for diagnosing diseases and pathologies of other fish species. However, before accepting the MS for publication, the authors should improve the MS, in particular check the English language.
Response: We would like to thank our reviewer for careful reading of the manuscript and also for the constructive comments. We thoroughly revised our manuscript and went through its different sections. The English of the manuscript was revised and ensured to be of academic quality; typos, grammatical errors, and sentences with vague meanings were fixed. Overlapped parts have been unified in the current version of the manuscript. Additional discussions and arguments have been added whenever indicated, as shown below in the detailed response.
My minor comments are below:
Comment: Line 19: Please delete: or C. Idella
Response: Thank you for your comment. This point has been corrected in the revised manuscript (line 19).
Comment: Line 20: Please to replace on “In the present study, different parts of the of the buccal cavity in C. idella were examined using light microscopy and scanning electron microscopy.”
Response: Thank you for your comment. This point has been corrected in the revised manuscript (line 20, 21).
Comment: Line 21: Requests to replace "We" on "They have been"
Response: Thank you for your comment. This point has been corrected in the revised manuscript (line 21, 22).
Comment: Line 32: Please delete: ”or C. idella”
Response: Thank you for your comment. This point has been corrected in the revised manuscript (line 32).
Comment: Line 47: Most of the keywords are repeated with the MS title. Please replace the repeated keywords with e.g.: mouth; vertebrates; digestive tract
Response: Thank you for your comment. This point has been corrected in the revised manuscript (line 48, 49). Now the keywords section reads as: Cyprinidae; digestive tract; goblet cell; jaw; lip; mouth; taste bud; teleost; tongue; ultrastructure; vertebrates.
Comment: Line: 79: Please delete: “or C. idella”Line
Response: Thank you for your comment. This point has been corrected in the revised manuscript (line 79).
Materials and Methods
Comment: Line 112: Please change “aquaria” to “tanks”
Response: Thank you for your comment. This point has been corrected in the revised manuscript (line 113).
Comment: Line 114: Please add: …..(MS222; E10521……
Response: Thank you for your comment. This point has been corrected in the revised manuscript (line 115).
Comment: Line 131: In what percentage of formalin were the samples preserved? Please add this information
Response: Thank you for your comment. Details about fixative concentration have been included in the revised manuscript (line 132).
Comment: Line 131: Were the samples first dehydrated in a series of alcohol, acetone and xylene? Please describe this in more detail
Response: Thank you for your comment. Details about sample processing for paraffin embedding have been included in the revised manuscript (line 132-134).
Comment: Line 132: How were the samples cut into sections? Using a microtome? Which one? Please provide the information
Response: Thank you for your comment. Details about sectioning of paraffin blocks have been included in the revised manuscript (line 134, 135).
Discussion
Comment: Line 486: Requests to replace “Contrary to what these studies show”
Response: Thank you for your comment. Unfortunately, this paragraph has been deleted upon an input from another reviewer.
Comment: Line 496: Requests to replace: "Consistent observation in this study"
Response: Thank you for your comment. This point has been considered in the revised manuscript (line 527).
Comment: Line 502: Please change “work” to “study”
Response: Thank you for your comment. Unfortunately, this sentence has been deleted upon an input from another reviewer.
Comment: Line 528: Please change “work” to “study”
Response: Thank you for your comment. This point has been considered in the revised manuscript (line 534).
Comment: Line 552: ’’Our’’ change to ’’ In this study”
Response: Thank you for your comment. This point has been considered in the revised manuscript (line 553).
Comment: Line 562: Change please “work” to “study”
Response: Thank you for your comment. This point has been considered in the revised manuscript (line 599).
Comment: Line 562: Please delete “our”
Response: Thank you for your comment. This point has been considered in the revised manuscript (line 599).
Comment: Line 582: Please change “work” to “study”
Response: Thank you for your comment. This point has been considered in the revised manuscript
Reviewer 2 Report
Comments and Suggestions for Authors
The article titled 'Spatiotemporal ultrastructural, histological, and morphometric changes in the buccal cavity of grass carp (Ctenopharyngodon idella) during fingerling, yearling, and adult stages' was very well written and presents interesting results. Before publication, I suggest only minor additions that do not affect the quality of the article:
- line 114 - please add the abbreviation of the agent used - MS222
- lines 489, 498-500 and others - where possible, please add English species names
Author Response
Comment:
The article titled 'Spatiotemporal ultrastructural, histological, and morphometric changes in the buccal cavity of grass carp (Ctenopharyngodon idella) during fingerling, yearling, and adult stages' was very well written and presents interesting results.
Response: We would like to thank our reviewer for careful reading of the manuscript and also for the constructive comments. We thoroughly revised our manuscript and went through its different sections. Our response to reviewer comments is detailed below:
Before publication, I suggest only minor additions that do not affect the quality of the article:
Comment: line 114 - please add the abbreviation of the agent used - MS222
Response: Thank you for your comment. The abbreviation of the used anesthetic agent was added, please see line 115 in the revised manuscript.
Comment: lines 489, 498-500 and others - where possible, please add English species names
Response: Thank you for your comment. English names of fish species were added, please see lines 74, 505, 506, 509, 510, 528, 530, 531, 550, 553, 583, 594 in the revised manuscript.
Reviewer 3 Report
Comments and Suggestions for Authors
In this paper, the authors make a detailed description of the epithelial lining of the buccal cavity of three age classes of the cypriniform Ctenopharyngodon idella based on SEM, histology and histochemistry. The text is generally smoothly written and the pictures are of good quality.
Yet, I have several general comments:
A large part of the results is an elaborate description of the surface of the oral mucosa in the three age classes by means of SEM, and this result is also prominent in the abstract. To the reader, it would be helpful to provide a table summarizing these SEM findings, so that differences between anatomical structures and between age classes can be easily tracked. Moreover, the discussion essentially relies on the histological and histochemical observations and counts, so it is unclear how the detailed SEM description of folds, ridges, grooves, furrows etc. contributes to our understanding of feeding strategy of this species, and, importantly, how describing the morphological changes throughout different life stages will aid to improve their health and productivity, as expressed in the aims.
The discussion furthermore suffers from an apparent ignorance on the basic anatomy of cypriniforms. Thus, the authors highlight (even in the abstract) the absence of teeth in the oral region, and the absence on the tongue, as if this is not long known (for all cypriniforms). In that sense, the whole passage (L483-493) should be removed or at least rewritten. Likewise, discussing the absence of teeth on the tongue as something the authors discover, should be removed.
The paragraph on the valve compares the structure with that of a hagfish, a jawless fish, which of course has a totally different mouth structure and prey capture system. This is not just ‘fish of different dietary behaviors’ (L505), but these are differences between agnathans and gnathostomes, and should be treated as such.
I therefore suggest to revise the discussion substantially, in accordance with the above remarks.
From a methodological perspective, the authors should indicate how they delimited regions when they counted goblet cells etc.. (e.g. the limit between ‘upper lip’ and ‘upper jaw’ in Fig. 8A is not obvious).
The reference list is highly biased towards literature that has been published after 2000 (one reference out of 42 is older than 2000). Histological and histochemical studies were very fashionable in the 1970s and 1980s, and the techniques have not changed. Thus, it is remarkable that none of these relevant papers is used/cited.
Specific comments:
L54: The tongue of ‘fish’: the hyoid arch-supported structure that is called the tongue is not homologous to the muscular organ in mammals
L63: ‘barbels’: no barbels in this species (see also Fig. 1)
L65: ‘toothless fishes’: not correct as term if species without oral dentition are meant. There are very few truly toothless teleost fish speies. Many clades have an oral and pharyngeal dentition, which differ however in function; trituration is often performed with pharyngeal rather than oral teeth.
L91-92: in what time span – weekly? daily?
L125: specify what ‘oral commissures’ are
L143: ‘osmic acid’: osmium tetroxide?
L178: ‘tong’ must be ‘tongue.’
L182: ‘to have consequent rostro-caudal alignment’: would one expect anything else?
L185: ‘intercellular ridges’: ‘inter’cellular means between cells, so how are these ridges supported (by what structure?)
L182,188: Please define taste buds type I or type II
L217: ‘distinct intercellular boundaries’: could these be thickened microridges close to the intercellular junction? (thus not ‘intercellular’?) – see also remark above
L240: what sort of structure are microtubercles, and how can they indicate high secretory activity?
L243: progressive changes: compared to what?
L347: the orientation of the valve is likely determined at the time of fixation, and therefore not relevant
L480: not clear – what is the relationship to the appearance, at three months, of a terminal mouth?
L549: what is the ‘metabranchial region’?
Fig. 1: In anatomical literature, the golden standard is to picture animals with head to the left.
Fig. 8G: there is a large structure rostral to the dentary bone, that is not labeled. Being quite large, it looks as if it supports the lower lip, and should therefore be considered in the description, or at least properly labeled on the figure. What is this?
In general, avoid the term ‘fish’ when meant as a general term: specify whether teleost, or actinopterygian, or perhaps chondrichthyan fish...
Use italics for genus and species names in the references.
The text would benefit from being checked by a native speaker for wording or syntax (e.g. L90: feeds on, L124: ‘extracted’: dissected out, excised?; L204: ‘concaved’ : concave is not a verb,..)
Author Response
Comment: In this paper, the authors make a detailed description of the epithelial lining of the buccal cavity of three age classes of the cypriniform Ctenopharyngodon idella based on SEM, histology and histochemistry. The text is generally smoothly written and the pictures are of good quality.
Response: We would like to thank our reviewer for careful reading of the manuscript and also for the constructive comments. We thoroughly revised our manuscript and went through its different sections. The English of the manuscript was revised and ensured to be of academic quality; typos, grammatical errors, and sentences with vague meanings were fixed. Overlapped parts have been unified in the current version of the manuscript. Additional discussions and arguments about the novelty of study findings have been added whenever indicated, as shown below in the detailed response.
Yet, I have several general comments:
Comment: A large part of the results is an elaborate description of the surface of the oral mucosa in the three age classes by means of SEM, and this result is also prominent in the abstract. To the reader, it would be helpful to provide a table summarizing these SEM findings, so that differences between anatomical structures and between age classes can be easily tracked. Moreover, the discussion essentially relies on the histological and histochemical observations and counts, so it is unclear how the detailed SEM description of folds, ridges, grooves, furrows etc. contributes to our understanding of feeding strategy of this species, and, importantly, how describing the morphological changes throughout different life stages will aid to improve their health and productivity, as expressed in the aims.
Response: Thank you for your comment. These points have been considered in the revised manuscript. A table summarizing all SEM findings was added to the manuscript as Table 1, Please see page 12 of the revised manuscript. We agree with our reviewer that the manuscript lacked in-depth discussions of the SEM findings of different components of the buccal cavity in the studied fish, please see lines 490-514 and 571-580 of the revised manuscript.
Comment: The discussion furthermore suffers from an apparent ignorance on the basic anatomy of cypriniforms. Thus, the authors highlight (even in the abstract) the absence of teeth in the oral region, and the absence on the tongue, as if this is not long known (for all cypriniforms). In that sense, the whole passage (L483-493) should be removed or at least rewritten. Likewise, discussing the absence of teeth on the tongue as something the authors discover, should be removed.
Response: Thank you for pointing this out. Based on our reviewer input, we have trimmed all basic information related to the buccal cavity of the studied species from the abstract and also from the discussion. Now the discussion focuses mainly on the SEM findings and their correlation to functional maturation of the buccal cavity.
Comment: The paragraph on the valve compares the structure with that of a hagfish, a jawless fish, which of course has a totally different mouth structure and prey capture system. This is not just ‘fish of different dietary behaviors’ (L505), but these are differences between agnathans and gnathostomes, and should be treated as such.
Response: Thank you for your comment. We apologize for such confusion. The indicated part has been deleted in the revised manuscript.
Comment: I therefore suggest to revise the discussion substantially, in accordance with the above remarks.
Response: We have made substantial changes to the discussion section based on comments of our reviewer.
Comment: From a methodological perspective, the authors should indicate how they delimited regions when they counted goblet cells etc.. (e.g. the limit between ‘upper lip’ and ‘upper jaw’ in Fig. 8A is not obvious).
Response: Thank you for your notice. Details about the extent of different parts of the buccal cavity of C. idella have added to the methodological section, please see lines 156-165 of the revised manuscript.
Comment: The reference list is highly biased towards literature that has been published after 2000 (one reference out of 42 is older than 2000). Histological and histochemical studies were very fashionable in the 1970s and 1980s, and the techniques have not changed. Thus, it is remarkable that none of these relevant papers is used/cited.
Response: We apologize for the unintended mistake. Several pieces of literature related to pioneering work on histology and histochemistry have been added to the bibliography section of the revised manuscript.
Specific comments:
Comment: L54: The tongue of ‘fish’: the hyoid arch-supported structure that is called the tongue is not homologous to the muscular organ in mammals
Response: Thank you for your comment. This point has been corrected in the revised manuscript, please see line 55, 56.
Comment: L63: ‘barbels’: no barbels in this species (see also Fig. 1)
Response: Thank you for your comment. This paragraph was meant to illustrate the diversity in the appearance of the buccal cavity among fish. This paragraph has been amended and rechecked for consistency of the meaning, please see lines 60-68.
Comment: L65: ‘toothless fishes’: not correct as term if species without oral dentition are meant. There are very few truly toothless teleost fish speies. Many clades have an oral and pharyngeal dentition, which differ however in function; trituration is often performed with pharyngeal rather than oral teeth.
Response: Thank you for your comment. This part has been corrected, please see lines 60-68.
Comment: L91-92: in what time span – weekly? daily?
Response: Thank you for your comment. This point has been corrected in the revised manuscript, please see line 92-94.
Comment: L125: specify what ‘oral commissures’ are
Response: Thank you for your comment. This point has been corrected in the revised manuscript, please see line 126.
Comment: L143: ‘osmic acid’: osmium tetroxide?
Response: Thank you for your comment. This point has been corrected in the revised manuscript, please see line 147.
Comment: L178: ‘tong’ must be ‘tongue.’
Response: Thank you for your comment. This point has been corrected in the revised manuscript, please see line 192.
Comment: L182: ‘to have consequent rostro-caudal alignment’: would one expect anything else?
Response: This part has been deleted.
Comment: L185: ‘intercellular ridges’: ‘inter’cellular means between cells, so how are these ridges supported (by what structure?)
Response: Explanations about microridges and how they are connected to the cytoskeletal elements of the superficial epithelial cells of the buccal mucosa have been added to the discussion, please see line 571-580.
Comment: L182,188: Please define taste buds type I or type II
Response: Definitions were added to the context, please see line 195-205.
Comment: L217: ‘distinct intercellular boundaries’: could these be thickened microridges close to the intercellular junction? (thus not ‘intercellular’?) – see also remark above
Response: The terminology used for this finding has been replaced with a more neutral one as follows: (lines 230-232) Regarding the lingual keratinocytes, they featured tightly packed surface microridges and were separated from each other by distinct ridge-like borders. This terminology has been used to replace all relevant descriptions throughout the manuscript.
Comment: L240: what sort of structure are microtubercles, and how can they indicate high secretory activity?
Response: Microtubercles could be linked to the process of exocytosis (PMID: 28491287). This point is stressed to some extent in the discussion, please line 512-514.
Comment: L243: progressive changes: compared to what?
Response: Thank you for your comment. This point has been corrected in the revised manuscript, please see line 257.
Comment: L347: the orientation of the valve is likely determined at the time of fixation, and therefore not relevant
Response: Thank you for your comment. This point has been corrected in the revised manuscript, please see line 367.
Comment: L480: not clear – what is the relationship to the appearance, at three months, of a terminal mouth?
Response: Thank you for your comment. To avoid confusion by the future readers, this part has been deleted.
Comment: L549: what is the ‘metabranchial region’?
Response: Thank you for your comment. This point has been corrected in the revised manuscript, please see line 387.
Comment: Fig. 1: In anatomical literature, the golden standard is to picture animals with head to the left.
Response: Thank you for your comment. Fish heads are now facing towards the left in the revised manuscript, please see page 5 of the revised manuscript.
Comment: Fig. 8G: there is a large structure rostral to the dentary bone, that is not labeled. Being quite large, it looks as if it supports the lower lip, and should therefore be considered in the description, or at least properly labeled on the figure. What is this?
Response: Thank you for your comment. The indicated part refers to the osteoblasts cluster of the dentary bone. This deduction is based on published work on zebrafish cranial bones (PMID: 27278890). Both Figure 8 and its legend were updated accordingly, please see pages 13 and 14 of the revised manuscript.
Comment: In general, avoid the term ‘fish’ when meant as a general term: specify whether teleost, or actinopterygian, or perhaps chondrichthyan fish...
Response: Thank you for your comment. Your advice has been followed and is much appreciated.
Comment: Use italics for genus and species names in the references.
Response: Thank you for your comment. Your advice has been followed and is much appreciated.